# Tuning of the Electrostatic Potentials on the Surface of the Sulfur Atom in Organic Molecules: Theoretical Design and Experimental Assessment

**DOI:** 10.3390/molecules28093919

**Published:** 2023-05-06

**Authors:** Ziyu Wang, Weizhou Wang, Hai-Bei Li

**Affiliations:** 1SDU-ANU Joint Science College, Shandong University, Weihai 264209, China; 2College of Chemistry and Chemical Engineering, Luoyang Normal University, Luoyang 471934, China; 3Marine College, Shandong University, Weihai 264209, China

**Keywords:** electrostatic potential, theoretical design, cocrystal structure, halogen bond, chalcogen bond

## Abstract

Noncovalent sulfur interactions are ubiquitous and play important roles in medicinal chemistry and organic optoelectronic materials. Quantum chemical calculations predicted that the electrostatic potentials on the surface of the sulfur atom in organic molecules could be tuned through the through-space effects of suitable substituents. This makes it possible to design different types of noncovalent sulfur interactions. The theoretical design was further confirmed by X-ray crystallographic experiments. The sulfur atom acts as the halogen atom acceptor to form the halogen bond in the cocrystal between 2,5-bis(2-pyridyl)-1,3,4-thiadiazole and 1,4-diiodotetrafluorobenzene, whereas it acts as the chalcogen atom donor to form the chalcogen bond in the cocrystal between 2,5-bis(3-pyridyl)-1,3,4-thiadiazole and 1,3,5-trifluoro-2,4,6-triiodobenzene.

## 1. Introduction

As one of the most important heteroatoms, the sulfur (S) atom is widely existent not only in the field of medicinal chemistry but also in the field of organic optoelectronic materials [1,2,3,4,5,6,7,8]. The well-known S-containing drugs are penicillin, sulfonamides, sulfones and thioethers. Among the top 200 small molecule pharmaceuticals by retail sales in 2020, 63 pharmaceuticals contain the S element [1]. Pfizer’s Paxlovid (Ritonavir-boosted Nirmatrelvir) is currently used in COVID-19 treatment. Ritonavir is also a S-containing compound. Organic optoelectronic materials have been a hot research topic for a long time. Many small-molecule/polymer organic semiconductors also contain the chemical element S [6,7,8]. To better understand the structures and functions of S-containing drugs and organic optoelectronic materials, it is important to scrutinize the roles of noncovalent S interactions. There have been two seminal papers which highlight the key roles of noncovalent S interactions: Meanwell and coworkers conducted a comprehensive and general survey of the roles of intramolecular and intermolecular noncovalent S interactions in drug design, organic synthesis and protein structure and function [5]; Huang and coworkers reviewed the organic and polymeric semiconductors enhanced by intramolecular noncovalent S interactions [8].

In some noncovalent S interactions, the S atoms are nucleophilic and act as the electron donors. In a recent account, Biswal and coworkers highlighted the roles of hydrogen bonds in which the S atoms act as the electron donors in molecular assemblies, structural biology and functional materials [9]. In some other noncovalent S interactions, the S atoms are electrophilic and act as the electron acceptors. Using electrostatic potentials, Burling and Goldstein identified that the S atoms in thiazole nucleosides could interact via positive sites with electron-rich oxygen atoms [10]. In a recent paper, Wojtkowiak et al. explored the role of the S···O chalcogen bond in stabilizing ligand conformation in the binding pocket of the carbonic anhydrase IX receptor mimic [11]. Scheiner reviewed the participation of S in both the hydrogen and chalcogen bonds [12]. The original paper showing that the S atoms can have positive potentials associated with σ-holes was published in 2007 [13]. Then, in 2008 a paper also by Murray et al. focused on σ-hole bonding by S-containing heterocycles [14]. The molecular electrostatic potential has been applied to the study of noncovalent interactions for a long time [15,16]. From the point of view of supramolecular chemistry, the strategy that tunes the electrostatic potentials on the surface of the S atom can be used to manipulate the noncovalent S interaction. The nucleophilic S atom can be tuned into an electrophilic S atom and vice versa by the through-bond (inductive) or through-space (field) effects of suitable substituents. Normally, the S-containing pharmacophores of drugs and S-containing central cores of organic optoelectronic materials should stay unchanged. Therefore, it is reasonable to consider the through-space effects of substituents other than the through-bond effects of substituents in precise tuning of the electrostatic potentials on the surface of the S atoms. In this work, the through-space effects of pyridyl substituents on the electrostatic potentials on the surfaces of S atoms were explored by selecting the organic compounds 1,3,4-thiadiazole (TD), 2,5-bis(2-pyridyl)-1,3,4-thiadiazole (2-BPTD) and 2,5-bis(3-pyridyl)-1,3,4-thiadiazole (3-BPTD) as model molecules (see Figure 1). Further, 1,4-diiodotetrafluorobenzene (1,4-DITFB) and 1,3,5-trifluoro-2,4,6-triiodobenzene (1,3,5-TFTIB) shown in Figure 1 were selected as probes to identify the interactive behavior of the S atoms by X-ray crystallographic experiments. Ideally, the positive σ-hole of the I atom in 1,4-DITFB or 1,3,5-TFTIB can attract the nucleophilic S atom to form the halogen bond [17,18,19], and the nucleophilic F atom in 1,4-DITFB or 1,3,5-TFTIB can attract the electrophilic S atom to form the chalcogen bond [20,21].

In this study, we only investigated the effect of pyridyl substituents on the electrostatic potentials on the surface of the S atom. Evidently, both the theoretical and experimental results will be very similar if we use other similar substituents such as furyl, thienyl, pyrimidinyl, etc. On the other hand, such a strategy can also be used to tune the electrostatic potentials on the surfaces of the other atoms.

## 2. Results

### 2.1. Theoretical Design

The molecular electrostatic potential maps of 2-BPTD, TD, 3-BPTD, 1,4-DITFB and 1,3,5-TFTIB are shown in Figure 2. The molecules 2-BPTD, TD and 3-BPTD have *C*_2V_ symmetry. The minimum electrostatic potential on the surface of the N atom of the pyridyl substituent in 2-BPTD is −27 kcal/mol, and the electrostatic potential on the surface of the S atom along the *C*_2_ axis of symmetry in 2-BPTD is −8 kcal/mol. The S atom in TD has two positive σ-holes and the maximum electrostatic potential on each σ-hole is +30 kcal/mol. Different from the case in 2-BPTD, the electrostatic potential on the surface of the S atom along the *C*_2_ axis of symmetry in TD becomes positive and is +11 kcal/mol. Evidently, it is the introduction of the pyridyl substituent that changes the electrophilic S atom into a nucleophilic S atom. In the molecule 3-BPTD, the maximum electrostatic potential on the surface of the H atom of the pyridyl substituent is +30 kcal/mol and the electrostatic potential on the surface of the S atom along the *C*_2_ axis of symmetry is +18 kcal/mol. In contrast to the corresponding value in TD, the electrostatic potential on the surface of the S atom along the *C*_2_ axis of symmetry in 3-BPTD becomes more positive, which means that the introduction of the pyridyl substituent can also make the electrophilic S atom more electrophilic.

The electron density transfer associated with the formation of intramolecular noncovalent bonds was explored by natural bond orbital (NBO) methods [22]. In the NBO theory, the second-order perturbation stabilization energy can be used to quantitatively evaluate the electron density transfer from the donor orbital to the acceptor orbital [22]. In the molecule 2-BPTD, the NBO analysis shows that there is a slight electron density transfer from the lone electron pair of the N atom in the pyridyl substituent to the σ* antibonding orbital of the S–C bond in the 1,3,4-thiadiazole ring. The second-order perturbation stabilization energy of this donor–acceptor orbital interaction is only 0.93 kcal/mol. In the molecule 3-BPTD, NBO analysis shows that there is no electron density transfer from the lone electron pair of the S atom in the 1,3,4-thiadiazole ring to the σ* antibonding orbital of the adjacent C–H bond in the pyridyl substituent. These results indicate that the effects of the intramolecular noncovalent bonds on the electrostatic potentials on the surface of the S atom are not large. Employing the additive electrostatic potential model proposed by Wheeler and Houk [23], we also calculated the additive electrostatic potentials (TD + 2pyridine) for 2-BPTD and 3-BPTD, respectively, and plotted their additive electrostatic potential maps. There is no obvious difference between the true and additive electrostatic potential maps of 2-BPTD or 3-BPTD, which indicates that the through-space effects of pyridyl substituents tune the electrostatic potentials on the surfaces of the S atoms.

Figure 3 demonstrates the electron density contour maps on the molecular planes of 2-BPTD, TD and 3-BPTD. The electron densities in the regions near the blue arrows are almost the same for the three molecules 2-BPTD, TD and 3-BPTD. The results are consistent with previous finding that the changes in the electrostatic potentials do not necessarily indicate changes in electron densities [23]. Again, these results support the conclusion that the changes in the electrostatic potentials on the surfaces of the S atoms are caused by the through-space effects of pyridyl substituents.

The electrophilic or nucleophilic nature of the S atom can be distinguished by using 1,4-DITFB or 1,3,5-TFTIB as a probe. The molecular electrostatic potential maps of 1,4-DITFB and 1,3,5-TFTIB are shown in Figure 2. Each of the I atoms in the two molecules has a positive σ-hole with a maximum electrostatic potential of about +30 kcal/mol. The F atoms in 1,4-DITFB and 1,3,5-TFTIB have negative electrostatic potentials. The maximum electrostatic potential on the F atom is in the negative σ-hole region and has a value of −7 kcal/mol. Note that the three lone-electron pairs on the F atom form a belt region of negative electrostatic potential, and the electrostatic potentials in this region are more negative than those in the negative σ-hole region. The I atom with a positive σ-hole can detect the nucleophilic S atom by the formation of the halogen bond, and the nucleophilic F atom can detect the electrophilic S atom by the formation of the chalcogen bond. At present, the detection of the halogen bond and chalcogen bond in the gas phase or in the liquid phase is still relatively difficult, while such detections can be readily made by X-ray crystallographic experiments.

### 2.2. Crystallographic Assessment

Two cocrystals were successfully solved in this study: cocrystal **1** was formed between 2-BPTD and 1,4-DITFB and cocrystal **2** was formed between 3-BPTD and 1,3,5-TFTIB. Figure 4 shows the crystal structures of **1** and **2**. The main crystal and structure refinement data for **1** and **2** are summarized in Table 1. It can be clearly seen from Figure 4 that there are a large number of π···π stacking interactions in the crystal structures of **1** and **2**. The 2D layered structures in **1** are formed by the π···π stacking interactions between 2-BPTD molecules and the π···π stacking interactions between 2-BPTD and 1,4-DITFB. The 2D layered structures in **2** are formed by the π···π stacking interactions between 3-BPTD molecules and the π···π stacking interactions between 3-BPTD and 1,3,5-TFTIB. Different 2D layered structures are linked together by the other intermolecular nonbonding interactions such as the hydrogen bonds, halogen bonds and chalcogen bonds to form the 3D structures of **1** and **2**.

Figure 5 shows the interaction energies of all the intermolecular nonbonding interactions in the crystal structure of **1**. These intermolecular nonbonding interactions include the π···π stacking interaction between 2-BPTD and 1,4-DITFB, π···π stacking interaction between 2-BPTD and 2-BPTD, C–H···N hydrogen bond, C–H···F hydrogen bond, C–H···I hydrogen bond, bifurcated C–I···(N···S) halogen bond and bifurcated C–I···(N–N) halogen bond. The existence of the hydrogen bonds and halogen bonds was determined according to their definitions from the International Union of Pure and Applied Chemistry (IUPAC) [19,24]. Firstly, the interactions are attractive, and secondly the attractive interactions occur between electrophilic halogen/hydrogen atoms and electron donors.

As can be seen in Figure 5, there is a net attractive interaction with the interaction energy of −0.71 kcal/mol between the I atom with a positive σ-hole and the nucleophilic S atom, which is consistent with our theoretical prediction that the electrostatic potentials on the surface of the S atom in 2-BPTD are negative. The interatomic distance between I and S in the monofurcated C–I···S halogen bond is 3.67 Å, which is less than the sum of their van der Waals radii (3.78 Å) [25]. This is another indicator of the formation of the monofurcated C–I···S halogen bond. In this study, the interaction energies of intermolecular nonbonding interactions were calculated with the conventional supermolecule method. It is very simple to calculate the interaction energy of the bifurcated C–I···(N···S) halogen bond (see Figure 6). However, it is a little challenging to separate the interaction energy of the monofurcated C–I···S halogen bond from the interaction energy of the bifurcated C–I···(N···S) halogen bond. In Figure 6, the geometries of the dimer between 3,5-dichloropyridine and 1,4-DITFB and the dimer between pyridine and 1,4-DITFB are kept the same as in the dimer between 2-BPTD and 1,4-DITFB, except that the positions of the saturated H atoms and Cl atoms are optimized (the Cartesian atomic coordinates for these dimers have been given in the Appendix A). Even so, we still cannot use the model dimer between pyridine and 1,4-DITFB to estimate the interaction energy of the monofurcated C–I···N halogen bond in the bifurcated C–I···(N···S) halogen bond because the electrostatic potentials on the surface of the N atom in pyridine are quite different from the corresponding ones in 2-BPTD. It can be clearly seen from Figure 6a,b that the electrostatic potentials on the surface of the N atom in 3,5-dichloropyridine are almost the same as the corresponding ones in 2-BPTD. In addition, NBO analyses show that the second-order perturbation stabilization energy of the interaction between the lone electron pair on the N atom and the C–I σ* antibonding orbital in the dimer between 2-BPTD and 1,4-DITFB is also almost the same as the corresponding one in the dimer between 3,5-dichloropyridine and 1,4-DITFB. Therefore, we can use the interaction energy of the dimer between 3,5-dichloropyridine and 1,4-DITFB to represent the strength of the monofurcated C–I···N halogen bond in the bifurcated C–I···(N···S) halogen bond. Finally, the interaction energy of the monofurcated C–I···S halogen bond can be calculated as the difference between the interaction energy of the bifurcated C–I···(N···S) halogen bond and the interaction energy of the monofurcated C–I···N halogen bond, and is estimated to be −0.71 kcal/mol. Although the monofurcated C–I···S halogen bond is not very strong, its attractive nature is enough to determine the crystal structure of **1**.

All the intermolecular nonbonding interactions in the crystal structure of cocrystal **2** and their interaction energies are shown in Figure 7. The strongest intermolecular nonbonding interaction is the π···π stacking interaction between 3-BPTD and 1,3,5-TFTIB with an interaction energy of −10.51 kcal/mol. The C–I···N halogen bond is much stronger than the bifurcated C–I···(N–N) halogen bond, whereas the bifurcated (C–H)_2_···F hydrogen bond is stronger than the C–H···F hydrogen bond. The weakest noncovalent interaction is the C–S···F chalcogen bond with an interaction energy of −0.53 kcal/mol. Again, the existence of the hydrogen bond, halogen bond and chalcogen bond in the crystal structure of **2** is judged according to their IUPAC definitions [19,21,24]. On the other hand, the interatomic distance between S and F in the C–S···F chalcogen bond is 3.02 Å, which is less than the sum of their van der Waals radii (3.27 Å) [25]. The much smaller S···F interatomic distance indicates the existence of the C–S···F chalcogen bond once again. The formation of the attractive C–S···F chalcogen bond proves that the S atom in 3-BPTD is electrophilic and the electrostatic potentials on the surface of the S atom are positive, which is also in agreement with our theoretical prediction. Here, we want to stress that, although the C–S···F chalcogen bond is weak like the C–I···S halogen bond in **1**, it still plays a significant role for the formation of **2**. The C–F bond could not point to the S atom if the S···F contact is repulsive.

## 3. Materials and Methods

### 3.1. Computational Methods

The geometries of the monomers TD, 2-BPTD, 3-BPTD, 1,4-DITFB and 1,3,5-TFTIB were fully optimized at the M06-2X/def2-TZVPP level of theory [26,27]. The subsequent frequency calculation showed that the structures shown in Figure 1 all corresponded to true minima on the potential energy surfaces. The Cartesian atomic coordinates for these structures have been given in the Appendix A. The electrostatic potential maps on the 0.001 a.u. electron density isosurfaces of the monomers were generated with the GaussView software [28]. The values of electrostatic potentials at the sites of interest were calculated by employing the Multiwfn program [29]. The interaction energies of the dimer were calculated with the supermolecule method at the PBE0-D3(BJ)/def2-TZVPP theory level [30,31,32]. The PBE0-D3(BJ)/def2-TZVPP calculations perform very well for the study of weakly bound complexes [33,34]. Correction of the basis set superposition error in interaction energy has been made by using the counterpoise method [35]. The geometries of the dimers were taken from the crystal structure, and only the hydrogen atom positions were optimized at the PBE0-D3(BJ)/def2-TZVPP theory level. It is well known that the position of the hydrogen atom cannot be located very accurately using X-ray diffraction. Other researchers also used a similar methodology for theoretical calculations of intermolecular nonbonding interactions in crystal structures [36,37,38,39,40]. The NBO analyses were carried out for both monomers and dimers by employing the PBE0-D3(BJ)/def2-TZVPP optimized geometries and densities [22,41]. All the density functional theory calculations and NBO analyses were performed with the GAUSSIAN 16 software suite [42].

### 3.2. Experimental Methods

The chemicals 2-BPTD, 3-BPTD, 1,4-DITFB and 1,3,5-TFTIB were purchased from either J&K Scientific (Beijing, China) or Sigma-Aldrich (St. Louis, MO, USA). The purities of these chemicals were higher than 98.0%. All these chemicals were used as received. The 3-BPTD has both cis and trans isomers [43,44]. As shown in Figure 1, we only considered the cis isomer of 3-BPTD in this work. The solvent chloroform was reagent grade and used without further purification. A series of cocrystallization reactions between 2-BPTD/3-BPTD and 1,4-DITFB/1,3,5-TFTIB were carried out by slow evaporation of the chloroform solutions of two reactants in 2:1, 1:1 and 1:2 molar ratios, respectively, at room temperature. Finally, only the single crystals of 1:1 cocrystal **1** between 2-BPTD and 1,4-DITFB and 1:1 cocrystal **2** between 3-BPTD and 1,3,5-TFTIB were obtained.

The single crystal X-ray diffraction data of **1** and **2** were collected at room temperature by using the SuperNova Rigaku Oxford Diffraction diffractometers with monochromated Mo-Kα radiation (λ = 0.71073 Å). The crystal structures were solved by direct methods and the SHELX-2014 program was used for the least-squares refinement [45]. Empirical absorption correction using spherical harmonics, implemented in the SCALE3 ABSPACK scaling algorithm, was applied in the CrysAlisPro program [46]. The hydrogen atoms of 2-BPTD and 3-BPTD were refined at idealized positions riding on the carbon atoms, with isotropic displacement parameters *U*_iso_(H) = 1.2*U*_eq_(C) and *d*(C–H) = 0.93 Å. The crystallographic information files of **1** and **2** have been deposited in the Cambridge Crystallographic Data Centre (CCDC) and can be downloaded free of charge. The CCDC deposition number for **1** is 2,162,326 and the CCDC deposition number for **2** is 2,162,328. At the same time, the crystallographic information files of **1** and **2** have also been provided in the Appendix A. The checkcif files for the two cocrystal structures can be found in the Appendix A.

## 4. Conclusions

The tuning of the electrostatic potentials on the surface of the S atom in organic molecules has been investigated by using a combined theoretical and experimental approach. The calculated results show that the introduction of the pyridyl substituents into 1,3,4-thiadiazole can change the electrostatic potentials on the surface of the S atom from positive to negative or from positive to much more positive. NBO analysis indicates that the effects of the intramolecular noncovalent bonds on the electrostatic potentials on the surface of the S atom are not large. The true and additive electrostatic potential maps of 2-BPTD or 3-BPTD are very similar, which means that it is the through-space effect of the pyridyl substituent that tunes the electrostatic potentials on the surface of the S atom. The results of the theoretical designs were all confirmed by the X-ray crystallographic experiments. The formation of the C–I···S halogen bond in the crystal structure of **1** confirms that the electrostatic potentials on the surface of the S atom in 2-BPTD are negative. The formation of the C–S···F chalcogen bond in the crystal structure of **2** confirms that the electrostatic potentials on the surface of the S atom in 3-BPTD are positive.

In summary, we have demonstrated the successful manipulation of noncovalent S interactions by the precise tuning of electrostatic potentials on the surface of the S atom. S is a very important element in the fields of medicinal chemistry and organic optoelectronic materials. Therefore, the results presented in this paper will open new dimensions in the design and synthesis of new drugs and organic optoelectronic materials.

## Figures and Tables

**Figure 1 molecules-28-03919-f001:**
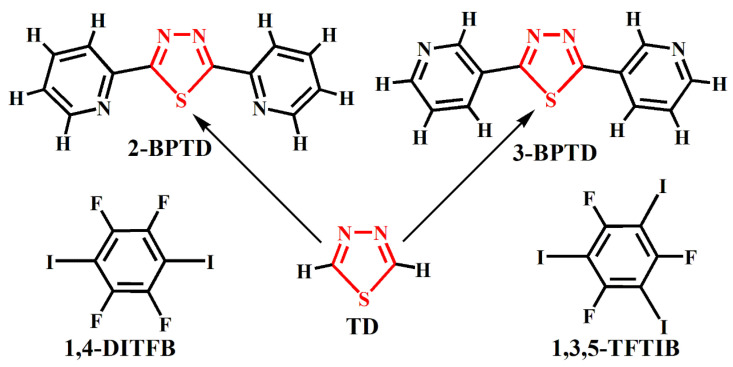
The molecular structures of 2-BPTD, 3-BPTD, TD, 1,4-DITFB and 1,3,5-TFTIB.

**Figure 2 molecules-28-03919-f002:**
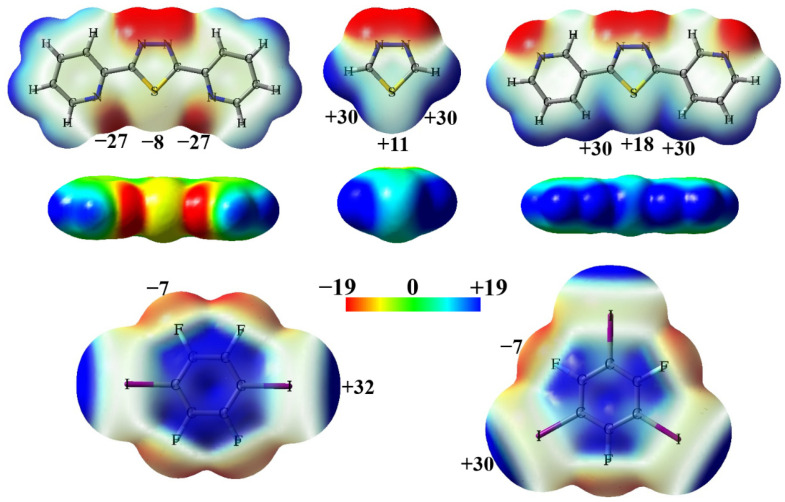
The front views of the transparent electrostatic potential maps of 2-BPTD, TD, 3-BPTD, 1,4-DITFB and 1,3,5-TFTIB, and the side views of the solid electrostatic potential maps of 2-BPTD, TD and 3-BPTD. The numbers (in kcal/mol) are the electrostatic potentials at the sites of interest.

**Figure 3 molecules-28-03919-f003:**
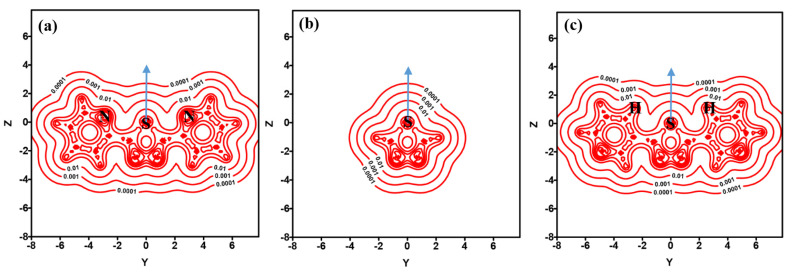
The electron density (in au) contour maps on the molecular planes of 2-BPTD (**a**), TD (**b**) and 3-BPTD (**c**). For clarity purpose, only the atoms of interest are marked in the figure. The distance along the Y or Z axis is in Å.

**Figure 4 molecules-28-03919-f004:**
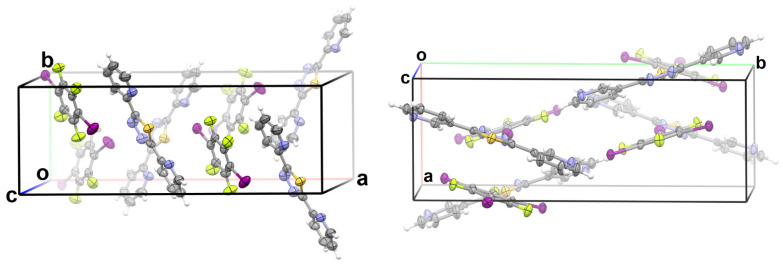
The unit cells for **1** (**left**) and **2** (**right**). The short contacts are not shown for the sake of clarity. Color code: H, white; C, gray; N, blue; F, yellow green; S, yellow; I, purple.

**Figure 5 molecules-28-03919-f005:**
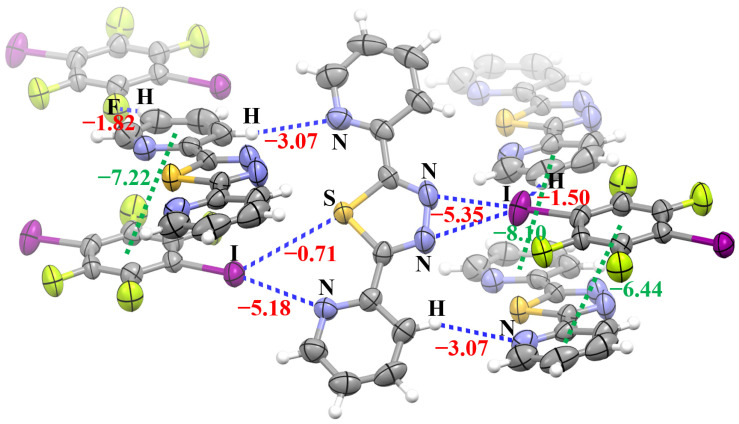
The interaction energies (red and green numbers, in kcal/mol) of the intermolecular nonbonding interactions in the crystal structure of **1**. The green numbers denote the π···π stacking interaction energies. Color code: H, white; C, gray; N, blue; F, yellow green; S, yellow; I, purple.

**Figure 6 molecules-28-03919-f006:**
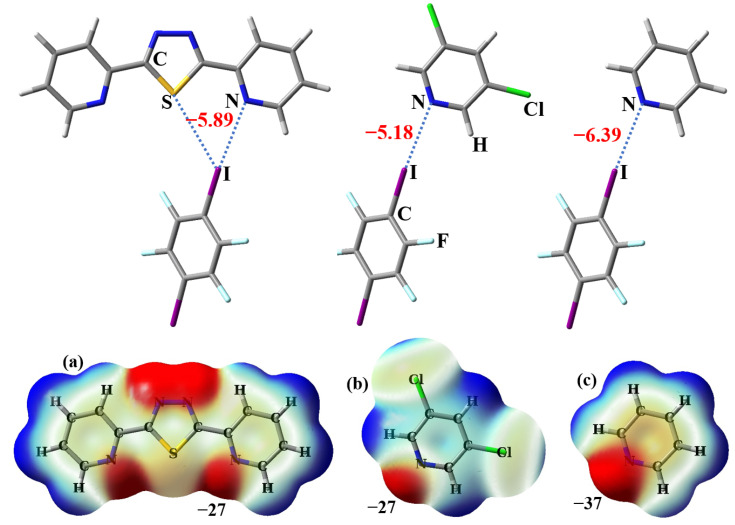
The interaction energies (in kcal/mol) of one bifurcated halogen bond and two monofurcated halogen bonds. Shown in the second row are the front views of the transparent electrostatic potential maps of 2-BPTD (**a**), 3,5-dichloropyridine (**b**) and pyridine (**c**) along with the electrostatic potential minima at the N atoms.

**Figure 7 molecules-28-03919-f007:**
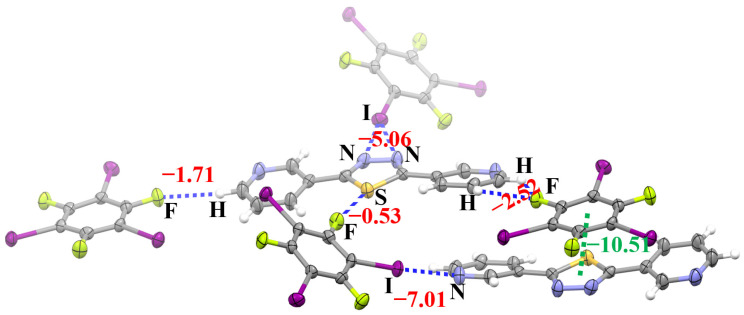
The interaction energies (red and green numbers, in kcal/mol) of the noncovalent interactions in the crystal structure of **2**. The green number denotes the π···π stacking interaction energy. Color code: H, white; C, gray; N, blue; F, yellow green; S, yellow; I, purple.

**Table 1 molecules-28-03919-t001:** Crystal and structure refinement data for **1** and **2**.

	1	2
CCDC deposition number	2,162,326	2,162,328
Empirical formula	C_18_H_8_F_4_I_2_N_4_S	C_18_H_8_F_3_I_3_N_4_S
Formula weight	642.14	750.04
Temperature/K	293 (2)	293 (2)
Crystal system	monoclinic	monoclinic
Space group	*Cc*	*P*2_1_/*n*
*a*/Å	22.0520 (5)	7.5248 (2)
*b*/Å	7.84693 (17)	19.6708 (5)
*c*/Å	11.8297 (2)	14.6179 (5)
*β*/°	99.301 (2)	103.947 (3)
Volume/Å^3^	2020.10 (8)	2099.94 (11)
*Z*	4	4
*ρ*_calc_/g‧cm^−3^	2.111	2.372
Color	colorless	colorless
Crystal size/mm^3^	0.29 × 0.25 × 0.21	0.31 × 0.28 × 0.12
Reflections collected	20,134	11,182
Independent reflections	3542	3854
*R* _int_	0.033	0.042
Number of refined parameters	263	263
Goodness-of-fit on *F*^2^	1.074	1.047
Final *R*_1_ index [I ≥ 2*σ*(I)]	0.0224	0.0327
Final *wR*_2_ index [I ≥ 2*σ*(I)]	0.0521	0.0668
Final *R*_1_ index [all data]	0.0235	0.0394
Final *wR*_2_ index [all data]	0.0529	0.0699

## Data Availability

The crystallographic data have been deposited in the Cambridge Structural Database as entries no. 2162326 and 2162328. The other data relevant to this article are contained within the article itself and in the Appendix A.

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
