# Peer review of "Tuning of the Electrostatic Potentials on the Surface of the Sulfur Atom in Organic Molecules: Theoretical Design and Experimental Assessment"

_molecules, 2023, doi:10.3390/molecules28093919_

Round 1

Reviewer 1 Report

The work is very interesting as it discusses noncovalent interactions. Such interactions are critical to keep the 3-D structure of large molecules,  heavily influence drug design, design of materials  and crystallinity, and in general synthesis of organic molecules. Therefore, scientific works analyzing this subject are very welcome.

The authors use basis set and functionals adequate for the systems investigated, the discussion, citations, and general presentation of the manuscript are good. However, there are some clarifications needed throughout the text. I would like to call the attention of the authors to the following.

1.     In the title of the manuscript, I would use Organic Molecules instead of Organic Molecule

2.     It is missing in the text information of NBOs. For example, in page 4 lines 103 and 104 it is written: The  second-order perturbation stabilization energy of this donor-acceptor orbital interaction is only 0.93 kcal/mol.  Where did this number come from? Similarly on page 7, lines  194-196 “In addition, NBO analyses show that the second-order perturbation  stabilization energy of the interaction between the lone electron pair on the N atom and  the C–I σ* antibonding orbital in the dimer between 2-BPTD and 1,4-DITFB is also almost  the same as the corresponding one in the dimer between 3,5-dichloropyridine and 1,4-197 DITFB” and on page 8, lines 244 and 245,  “The NBO analyses were carried  out for both monomers and dimers [40], no further information is provided about NBO. Information on how the NBOs were calculated (Computational Methods)  and a picture of them must be provided where their calculation is mentioned (Discussion).

3.     On page 7 lines, 192-193, “It can be clearly seen from Figure 6 that the electrostatic potentials on  the surface of the N atom in 3,5-dichloropyridine are almost the same as the corresponding ones in 2-BPTD”, this statement is not quite correct if one checks the values. Please explain it.

4.     Figures 3, 6, and 7 must be identified also by items a, and c to facilitate the reading.

5.     The English must be reviewed. There are long paragraphs, beyond mistypes such as “existences” (page 5, line 161 and page 8, line 214) instead existence.

Author Response

The work is very interesting as it discusses noncovalent interactions. Such interactions are critical to keep the 3-D structure of large molecules, heavily influence drug design, design of materials and crystallinity, and in general synthesis of organic molecules. Therefore, scientific works analyzing this subject are very welcome.

The authors use basis set and functionals adequate for the systems investigated, the discussion, citations, and general presentation of the manuscript are good. However, there are some clarifications needed throughout the text. I would like to call the attention of the authors to the following.

  1. In the title of the manuscript, I would use Organic Molecules instead of Organic Molecule

Reply: In the title of the manuscript, “Organic Molecule” has been changed into “Organic Molecules”.

  1. It is missing in the text information of NBOs. For example, in page 4 lines 103 and 104 it is written: The second-order perturbation stabilization energy of this donor-acceptor orbital interaction is only 0.93 kcal/mol. Where did this number come from? Similarly on page 7, lines 194-196 “In addition, NBO analyses show that the second-order perturbation stabilization energy of the interaction between the lone electron pair on the N atom and the C–I σ* antibonding orbital in the dimer between 2-BPTD and 1,4-DITFB is also almost the same as the corresponding one in the dimer between 3,5-dichloropyridine and 1,4-197 DITFB” and on page 8, lines 244 and 245, “The NBO analyses were carried out for both monomers and dimers [40], no further information is provided about NBO. Information on how the NBOs were calculated (Computational Methods) and a picture of them must be provided where their calculation is mentioned (Discussion).

Reply: We thank the reviewer very much for the helpful suggestions.

On page 4, the following sentences were added:

“The electron density transfer associated with the formation of the intramolecular noncovalent bonds was explored by the natural bond orbital (NBO) methods [22]. In the NBO theory, the second-order perturbation stabilization energy can be used to quantitatively evaluate the electron density transfer from the donor orbital to the acceptor orbital [22].”

On page 8, the following sentences were added:

“The NBO analyses were carried out for both monomers and dimers by employing the PBE0-D3(BJ)/def2-TZVPP optimized geometries and densities [22,41]. All the density functional theory calculations and NBO analyses were performed with the GAUSSIAN 16 software suite [41].”

  1. On page 7 lines, 192-193, “It can be clearly seen from Figure 6 that the electrostatic potentials on the surface of the N atom in 3,5-dichloropyridine are almost the same as the corresponding ones in 2-BPTD”, this statement is not quite correct if one checks the values. Please explain it.

Reply: We have changed “from Figure 6” into “from Figures 6(a) and 6(b)”. It can be clearly seen from Figures 6(a) and 6(b) that the values are the same.

  1. Figures 3, 6, and 7 must be identified also by items a, and c to facilitate the reading.

Reply: These figures have been revised according to the reviewer’s suggestion.

  1. The English must be reviewed. There are long paragraphs, beyond mistypes such as “existences” (page 5, line 161 and page 8, line 214) instead existence.

Reply: We thank the reviewer for pointing out these errors to us. These errors have been corrected, and the English of this manuscript has also been improved.

Reviewer 2 Report

Review on Molecules-2342883 (crystallographic part only)           

Crystal structure determinations and refinements are mostly sound (see below), although unfortunately only limited data sets were measured (low 2theta). For Mo-radiation 2theta should be at least 56 degree.

Discussion as well as presentation need to be very improved.

For space group Cc structure the use of TWIN and BASF commands is essential. Re-refine accordingly.

As intermolecular D-H…A contacts are discussed/interesting the use of HTAB command for both structures is recommended. Thus the relevant tables will be generated.

Also missing are some relevant geometric data (bond lengths and angles) and torsion angles for the BPTD moieties.

In general there is no need to write cocrystal 1 and cocrystal 2 throughout the manuscript: just 1 and 2 will suffice and shorten the text.

For 1 and 2 molecular structures must be shown with anisotropic displacement ellipsoids; Figure 4 is not really helpful. Provide labels for all atoms but hydrogens.

Table 1 states data collection temperatures of 287 and 293 K. You are sure of these different temperatures? The .INS files include no TEMP commands, so 293 is the default. Furthermore, surely neither 287 nor 293 K is constant throughout data collection. Temperatures must be given as 293(2) K. Obviously .CIF files have been edited – this is not serious.

There is no statement concerning (the necessary) absorption corrections (paragraph 4.2), additionally in Table 1 crystal sizes and number of refined parameters are missing.

Para 4.2, line 261 ff.  must also contain correct details of H-atom refinement (authors discuss H-bonds), e.g. “Hydrogen atoms were refined at idealized positions riding on the carbon atoms with isotropic displacement parameters Uiso(H) = 1.2Ueq(C) and C-H 0.93 A.”

More details:

Para 2.2, line 138 ff.  “solved” instead of “resolved”.

Ditto, line 142: delete complete sentence, no necessary information.

Ditto, line 143: delete “In Figure 4 … of clarity.” This information has to be given with the caption of Figure 4, instead. Must read “.. for the sake ..”

Ditto, line 150 ff. These interactions may well be interesting, so a corresponding table would be helpful. “Noncovalent” is the wrong term, “intermolecular nonbonding interactions” would be correct. The mentioned “halogen bonds” and “chalcogen bonds” are not those from halogen or sulphur atoms to their direct neighbors. Re-write.

Para 4.2, line 259: must read “…crystal structures were solved by …” (not “resolved”)

Table 1: delete alpha and gamma angles, this is trivial for monoclinic systems.

Author Response

Crystal structure determinations and refinements are mostly sound (see below), although unfortunately only limited data sets were measured (low 2theta). For Mo-radiation 2theta should be at least 56 degree.

Discussion as well as presentation need to be very improved.

Reply: We thank the reviewer for very helpful suggestions and comments. We have revised the manuscript according to the reviewer’s suggestions and comments. The discussion as well as presentation have also been improved.

In this revised version, the updated crystallographic information files of cocrystal 1 and cocrystal 2 have also been provided in the supplementary material.

For space group Cc structure the use of TWIN and BASF commands is essential. Re-refine accordingly.

Reply: The crystal structures have been refined using the TWIN and BASF commands.

As intermolecular D-H…A contacts are discussed/interesting the use of HTAB command for both structures is recommended. Thus the relevant tables will be generated.

Reply: We have used the HTAB command for both structures. However, there are no strong D-H…A contacts found in both structures. On the other hand, the strong halogen bonds cannot be identified by the software at present.

Also missing are some relevant geometric data (bond lengths and angles) and torsion angles for the BPTD moieties.

Reply: The crystallographic information files of cocrystal 1 and cocrystal 2 have also been updated.

In general there is no need to write cocrystal 1 and cocrystal 2 throughout the manuscript: just 1 and 2 will suffice and shorten the text.

Reply: We thank the reviewer for the suggestion and have deleted the word “cocrystal”.

For 1 and 2 molecular structures must be shown with anisotropic displacement ellipsoids; Figure 4 is not really helpful. Provide labels for all atoms but hydrogens.

Reply: Figures 1, 2, 3 and 6 were produced by the results from the quantum chemical calculations not from the crystal structures. According to the reviewer’s suggestion, we have changed Figures 5 and 7 into the ones with anisotropic displacement ellipsoids. For clarity, only the atoms involved in the noncovalent bonds were labeled.

Figure 4 was used to show the π···π stacking interactions in the two crystal structures.

Table 1 states data collection temperatures of 287 and 293 K. You are sure of these different temperatures? The .INS files include no TEMP commands, so 293 is the default. Furthermore, surely neither 287 nor 293 K is constant throughout data collection. Temperatures must be given as 293(2) K. Obviously .CIF files have been edited – this is not serious.

Reply: We thank the reviewer for very helpful suggestion. Temperatures have been given as 293(2) K.

We did not change any data in the .CIF files, and only added some authors’ information in order to eliminate the checkCIF publication errors.

There is no statement concerning (the necessary) absorption corrections (paragraph 4.2), additionally in Table 1 crystal sizes and number of refined parameters are missing.

Reply: The statement concerning absorption correction has been added in paragraph 4.2. The crystal sizes and number of refined parameters have been added in Table 1.

Para 4.2, line 261 ff. must also contain correct details of H-atom refinement (authors discuss H-bonds), e.g. “Hydrogen atoms were refined at idealized positions riding on the carbon atoms with isotropic displacement parameters Uiso(H) = 1.2Ueq(C) and C-H 0.93 A.”

Reply: We thank the reviewer for this suggestion, and have added the sentence “Hydrogen atoms were refined at idealized positions riding on the carbon atoms with isotropic displacement parameters Uiso(H) = 1.2Ueq(C) and C-H 0.93 Å”.

More details:

Para 2.2, line 138 ff. “solved” instead of “resolved”.

Reply: The word “resolved” has been changed into “solved”.

Ditto, line 142: delete complete sentence, no necessary information.

Reply: This sentence has been deleted.

Ditto, line 143: delete “In Figure 4 … of clarity.” This information has to be given with the caption of Figure 4, instead. Must read “.. for the sake ..”

Reply: We have deleted the sentence “In Figure 4 … of clarity.” This information has been given in the caption of Figure 4.

Ditto, line 150 ff. These interactions may well be interesting, so a corresponding table would be helpful. “Noncovalent” is the wrong term, “intermolecular nonbonding interactions” would be correct. The mentioned “halogen bonds” and “chalcogen bonds” are not those from halogen or sulphur atoms to their direct neighbors. Re-write.

Reply: We thank the reviewer for helpful suggestion. The term “noncovalent interactions” has been changed into “intermolecular nonbonding interactions”.

Para 4.2, line 259: must read “…crystal structures were solved by …” (not “resolved”)

Reply: The word “resolved” has been changed into “solved”.

Table 1: delete alpha and gamma angles, this is trivial for monoclinic systems.

Reply: We have deleted the alpha and gamma angles.

Round 2

Reviewer 2 Report

Crystallographic part is much better now.